# Novel Selection Approaches to Identify Antibodies Targeting Neoepitopes on the C5b6 Intermediate Complex to Inhibit Membrane Attack Complex Formation

**DOI:** 10.3390/antib10040039

**Published:** 2021-10-12

**Authors:** Lasse Stach, Emily K. H. Dinley, Nadia Tournier, Ryan P. Bingham, Darren A. Gormley, Jo L. Bramhall, Adam Taylor, Jane E. Clarkson, Katherine A. Welbeck, Claire L. Harris, Maria Feeney, Jane P. Hughes, Armin Sepp, Thil D. Batuwangala, Semra J. Kitchen, Eva-Maria Nichols

**Affiliations:** 1Biopharm Discovery, GlaxoSmithKline, Gunnels Wood Road, Stevenage SG1 2NY, UK; Lasse.x.Stach@gsk.com (L.S.); Emily.x.dinley@gsk.com (E.K.H.D.); Nadia.2.Tournier@gsk.com (N.T.); Jane.e.clarkson@gsk.com (J.E.C.); katherine.a.welbeck@gsk.com (K.A.W.); thil.d.batuwangala@gsk.com (T.D.B.); 2Screening, Profiling and Mechanistic Biology, GlaxoSmithKline, Gunnels Wood Road, Stevenage SG1 2NY, UK; Ryan.p.bingham@gsk.com; 3Immunology Research Unit, GlaxoSmithKline, Gunnels Wood Road, Stevenage SG1 2NY, UK; Darren.a.gormley@gsk.com (D.A.G.); Jo.l.bramhall@gsk.com (J.L.B.); Claire.Harris@ncl.ac.uk (C.L.H.); Maria.x.feeney@gsk.com (M.F.); j.hughes@gyroscopetx.com (J.P.H.); semra.j.kitchen@gsk.com (S.J.K.); 4Clinical Pharmacology & Experimental Medicine, GlaxoSmithKline, Gunnels Wood Road, Stevenage SG1 2NY, UK; Adam.taylor2@astrazeneca.com; 5Systems Modelling and Translational Biology, GlaxoSmithKline, Gunnels Wood Road, Stevenage SG1 2NY, UK; armin.z.sepp@gsk.com

**Keywords:** therapeutic antibody, complement, antibody discovery, neoepitope, terminal pathway

## Abstract

The terminal pathway of complement is implicated in the pathology of multiple diseases and its inhibition is, therefore, an attractive therapeutic proposition. The practicalities of inhibiting this pathway, however, are challenging, as highlighted by the very few molecules in the clinic. The proteins are highly abundant, and assembly is mediated by high-affinity protein–protein interactions. One strategy is to target neoepitopes that are present transiently and only exist on active or intermediate complexes but not on the abundant native proteins. Here, we describe an antibody discovery campaign that generated neoepitope-specific mAbs against the C5b6 complex, a stable intermediate complex in terminal complement complex assembly. We used a highly diverse yeast-based antibody library of fully human IgGs to screen against soluble C5b6 antigen and successfully identified C5b6 neoepitope-specific antibodies. These antibodies were diverse, showed good binding to C5b6, and inhibited membrane attack complex (MAC) formation in a solution-based assay. However, when tested in a more physiologically relevant membrane-based assay these antibodies failed to inhibit MAC formation. Our data highlight the feasibility of identifying neoepitope binding mAbs, but also the technical challenges associated with the identification of functionally relevant, neoepitope-specific inhibitors of the terminal pathway.

## 1. Introduction

The complement system forms an essential part of the innate immune response. Consisting of more than 30 serum and cell surface proteins, some of which are highly abundant, it functions as a first line of defense against infectious pathogens [1]. Once activated by one of three activation pathways, it stimulates phagocytosis by macrophages, releases pro-inflammatory signaling molecules and elicits direct lysis of pathogens and cells [2]. In addition, complement plays a role in the clearance of apoptotic cells by marking them for removal [3]. Complement activation requires tight regulation by many complement inhibitors for example, CD55 and CD59, which prevent damage to healthy tissue [4].

Mechanistic as well as genetic evidence has linked dysregulation of complement activation to several diseases. These include rare diseases such as Paroxysmal Nocturnal Hemoglobinuria (PNH), Atypical Hemolytic Uremic syndrome (aHUS) and Complement 3 Glomerulopathy (C3G), as well as common, chronic neurodegenerative pathologies like Alzheimer’s Disease [5,6]. Thus, there is considerable interest and opportunity in the complement system as a drug target [7,8].

Despite this, only a small number of complement drugs have been approved since the 2007 approval of eculizumab (Soliris^®^, Alexion Pharmaceuticals, Boston, MA, USA), indicative of challenges in targeting the complement system. The major challenge is the high abundance and fast turnover rates of some complement proteins necessitating high doses and frequent dosage regimens. For biological modalities, this is particularly challenging due to the high cost of goods. The success of the humanized anti-C5 monoclonal antibody eculizumab in treating PNH and aHUS provides validation of C5 as a drug target. However, it also highlights the challenges inherent in targeting complement proteins because these are highly abundant and have rapid turnover rates [9,10]. The maintenance dose for treatment of aHUS is a bi-weekly intravenous infusion of 1200 mg of Eculizumab. These high doses are one of the reasons for the high cost of eculizumab at more than £300,000 per annum in the UK [11]. 2013). More recently, (ravulizumab/Ultomiris^®^, Alexion Pharmaceuticals), an engineered variant of eculizumab that somewhat addresses the dosing challenges, has been approved.

One strategy to achieve lower dosing, which could improve patient access, is to develop therapeutics that specifically target neoepitopes exclusive to the less abundant activated forms of complement proteins. This could also address the challenges of target-mediated drug disposition reported for Eculizumab as well as “C3 bypass” cleavage of C5 which causes continued C5 cleavage in the presence of adequate anti-C5 inhibition [12,13]. Examples of complement neoepitope targeting therapeutics include the H17 antibody developed by Elusys, which specifically binds to activated C3 complement [14]. The H17 antibody binds to an epitope on the CUB domain of C3 which is accessible in C3b, but occluded in C3 [15]. Similarly, the S77 antibody developed by Genentech, also recognizes a C3b neoepitope located on the MG7 domain of C3 [16]. The IFX-1 antibody recognizes a neoepitope on one cleavage product of Complement C5, the anaphylatoxin C5a [17], which is further downstream in the complement cascade. This antibody has been shown to be efficacious as an anti-inflammatory in the treatment of African green monkeys infected with influenza A [18].

The membrane attack complex (MAC) is generated downstream of the complement protein C5 in the terminal cascade. Direct tissue damage by MAC is implicated in numerous pathologies with the mechanistically best-defined examples being PNH and aHUS. There is also increasing evidence to suggest that sub-lytic (i.e., non-pore forming) levels of MAC on the cell surface have numerous deleterious consequences on cell function [19,20]. Eculizumab inhibits cleavage of C5 and thus prevents MAC formation and C5a generation. C5a is a pro-inflammatory effector and it could be beneficial to selectively target formation of MAC, but without direct targeting of C5a. We therefore aimed to develop a MAC-specific inhibitor and set out to generate an anti-neoepitope mAb targeting C5b6, an intermediate complex in MAC formation (Figure 1A). Computational modelling supported the hypothesis that an antibody with an affinity of 1–2 pM would enable a favorable dosing regimen of monthly subcutaneous injections.

Given that the assembly of the terminal complement complex (TCC) is governed by high-affinity protein–protein interactions and involves large molecular surfaces, the system is likely to be more amenable to inhibition using biologicals rather than small molecules. Following cleavage of C5, the unstable C5b product binds C6 in a practically irreversible interaction, resulting in the formation of the stable C5b6 complex that forms the nucleus for TCC formation [21,22]. Figure 1B illustrates the comprehensive structural rearrangements undergone by C5b and C6 upon complex formation indicating the formation of neoepitopes.

The aim of this study was to generate a fully human IgG1, Fc-disabled monoclonal antibody that targets a neoepitope on the C5b6 complex of the TCC for potential use in disease indications in which terminal pathway effector function is known to drive pathology. Here, we describe the approach used to identify MAC-specific inhibitors targeting C5b6 neoepitopes.

## 2. Materials and Methods

### 2.1. Complement System Dose Modelling

All modelling and simulations were performed using MatLab SimBiology v5.2. C5b6 (The MathWorks, Inc., Natick, MA, USA) complex formation and proposed antibody binding were modelled using kinetic parameters from [23]. In addition, the effect of the faster C7 on-rate reported by Thai and Ogata, 2005 on target engagement was also modelled. Details are given in Appendix A [24].

### 2.2. Antibody Selections

Antibody clones were selected from Adimab LLC platform libraries or newly generated libraries with re-diversified CDRs, according to the protocols developed by Adimab LLC (Adimab, Lebanon, NH, USA). All antigens used were purchased from Complement Technology and were biotinylated prior to use via amine coupling. Magnetic bead selections were performed using streptavidin beads from Miltenyi (MACS^®^Miltenyi, Bergisch Gladbach, Germany) and FACS selections were performed on a BD ARIA II. Yeast populations were sorted based on binding to biotinylated C5b6 antigen, IgG expression, or a lack of binding to C5, C6 or a polyspecificity reagent (PSR) [25].

The initial selections consisted of two rounds of magnetic bead selections using 10 nM C5b6 and one round of FACS using 10 nM C5b6. The heavy chains of this round were then transformed into yeast containing a diversified light chain library from Adimab LLC. These libraries were then used to perform one round of MACS using 10 nM C5b6 and five rounds of FACS. The FACS rounds were as follows: 1: 10 nM C5b6, 2: 10 nM C5, 10 nM C6 and PSR, 3: 10 nM C5b6, 4: C7 competition, 5: 1 nM C5b6. The C7 competition was applied by binding 10 nM biotinylated C5b6 to the yeast to equilibrium, the yeast was then labelled with SA-633 (Streptavidin Alexa Fluor 633, Invitrogen, Waltham, MA, USA). To block any unbound biotin binding sites on the SA-633, biotin was used as a blocking agent post labelling. The yeast was then incubated with 100 nM biotinylated C7 for 20 min and labelled with EAPE (ExtrAvidin^®^−R-Phycoerythrin, Sigma Aldrich, St. Louis, MO, USA). Subsequently selecting clones that show high C5b6 binding and a lack of binding to C7. Round 4 was duplicated as above, except for using unlabeled C5b6. The plots in Figure 2B show the dual labelling of C5b6 and C7. The final round used the outputs from both arms of round 4. The final outputs were plated out on agar plates, and 95 colonies from each library output were picked and sequence verified. Unique sequences were expressed and purified as below.

For affinity maturation, new libraries re-diversified within the CDRH1 and CDRH2 regions were generated from eight selected clones. One round of MACS was performed followed by six rounds of FACS, using the following antigen concentrations: 1: 10 nM C5b6, 2: 10 nM C5, 10 nM C6 and PSR, 3: 10 nM C5b6, 4: C7 competition, 5: 1 nM C5b6, 6: 10 nM C5b6 with competition with 400 nM parental IgG. The C7 competition was performed by pre-incubating 10 nM unlabeled C5b6 with the cells, followed by incubation with 100 nM biotinylated C7 for 20 min, labelled with EAPE and subsequent FACS selection. The parental IgG competition was performed by the pre-incubation of 400 nM parental IgG with 10 nM C5b6 for 15 min prior to incubation with the cells. The gates for the final round were positioned to collect ~0.1% of the population that also, if possible, had better binding affinities than their corresponding parent. The final outputs were plated out on agar plates and 95 colonies from each library output were picked.

### 2.3. Antibody Expression and Purification

Antibody clones were expressed as fully human IgG1s from a proprietary yeast strain (Adimab) and purified using protein A affinity chromatography followed by buffer exchange into PBS. Fab fragments were generated by papain digestion and subsequent passage through a protein A affinity column to remove intact IgGs and Fc fragments. Introduction of Fc disabling mutation, such as of LAGA (L235A/G237A)/LALA (L234A/L235A), would have taken place later in the molecule discovery process [26].

### 2.4. Cloning, Expression and Purification of an Anti-C5 and Anti-MAC Tool Antibodies

The variable region sequences of an anti-C5 antibody were taken from European Patent EP 2359834, cloned into a pEF vector, expressed in CHO cells. This antibody has a reported affinity for C5 of <50 pM and prevents cleavage of C5 into C5a and C5b by the C5 convertase [27,28]. The anti-C7 mouse mAb was derived in-house from immunization of a transgenic mouse and generation of a hybridoma line with standard techniques. Both antibodies were purified by protein A affinity and size exclusion chromatography.

### 2.5. Biolayer Interferometry

A BLI assay was run on an Octet RED384 instrument (ForteBio, Fremont, QC, Canada). Antibodies were captured on protein A sensors (ForteBio). Following a buffer wash in PBSF, the sensors were dipped into an analyte solution for 180 s and the binding response at the end of the contact time recorded. For the screening of the naïve selection outputs, 50 nM of C5b6 (Comptech), C5 (Comptech) or C6 (Comptech) were used as analytes. For the screening of the affinity matured clones, 600 nM of C6 was used.

### 2.6. Terminal Complement Assay

The TCC assay was performed on a BioRobot FX. Normal human serum was diluted in 1× CFD buffer to 4% (*v/v*) and 25 C was added into polypropylene 96-well plates containing the anti-C5b6 antibodies and controls and incubated for 30 min on ice. The concentrations of test antibodies and controls are detailed in the Appendix A. Complement was activated with the addition of Zymosan (11 mg/mL), 5 μL per well, and incubated for 30 min at 37 °C. After incubation, complement activation was stopped using 11 μL of chilled EDTA (0.5 M) followed by 100 μL of chilled D:PBS to all the wells. The plates were centrifuged at 1000× *g* for 10 min at 4 °C. The supernatants were evaluated using the BD human C5b-9 ELISA Assay. The capture antibody provided within the kit was diluted 1:1000 with 0.1 M Carbonate-Bicarbonate buffer and added into ELISA Max high bind plates at 100 μL per well and incubated overnight at 4 °C. The plates were washed using a Biotek plate washer with PBS containing 0.05% Tween-20 and blocked for 1 h at room temperature with 10% heat inactivated fetal bovine serum in D:PBS at 100 μL per well. Working detection antibodies 1000× detector antibody and 1000× Streptavidin-HRP were combined in 10% heat inactivated fetal bovine serum in D:PBS. After blocking, the plates were washed with PBS with 0.05% Tween-20 and 100 μL of the prepared working detection antibodies was added and incubated for 1 h at room temperature. After incubation, the plates were washed with PBS with 0.05% Tween-20 and 100 μL of substrate solution TMB was added into each well and incubated for up to 30 min at room temperature in the dark. The reaction was stopped with 50 μL of 0.25 M HCl. The absorbance was read at 450 nm and 570 nm using a Biotek Epoch plate reader. The data were normalized against 0% inhibition.

### 2.7. Surface Plasmon Resonance

All Biacore experiments were performed using HBS-EP+ (Teknova) as a running buffer. An anti-C5 antibody was diluted to 50 μg/mL in 50 mM NaAc pH4.0 and amine coupled using a Biacore amine coupling kit to a Biacore CM5 chip on a Biacore 8K instrument (all GE Healthcare) according to the manufacturer’s instructions on all flow channels. For the MAC assembly assay, C5 (Quidel) or C5b6 (Comptech) were captured on the sample channels at 4 nM and nothing was captured on the reference channels. C7, C8 and C9 (all Comptech) were then flowed over all surfaces at concentrations of 4, 8, and 50 nM, respectively. For the kinetic run to determine the C7 on-rate, C5b6 was captured at 4 nM and C7 injected at 0, 1, 2 and 4 nM. For the kinetic analysis of the eight anti-C5b6 clones, C5b6 was captured at 4.5 nM and antibodies injected at the concentrations indicated in the figures. All interactions were fitted to a 1:1 model with local Rmax fitting. For the screening of the affinity matured clones, C5b6 or C5 were captured at 5 nM on a CM5 chip with immobilized anti-C5 antibody (immobilization as described above) on a Biacore4000 instrument (GE Healthcare) and antibody injected at 500 nM for 240 s. The binding response at the end of the contact time was used to plot the relative binding of each clone.

### 2.8. C9 Oligomerization HTRF Assay

A total of 0.5 mg of Complement C9 (Comptech) at 1.05 mg/mL was dialyzed into 100 mM sodium bicarbonate. A total of 50 ug of amine reactive fluorescein isothiocyanate (FITC) (Pierce) were dissolved in DMSO to 10 mg/mL. Then, 250 μL of the C9 solution reacted with 2.5 μL of the FITC solution, thus at a molar ratio of dye/protein of 2:1, for 2 h at room temperature. Free FITC was separated from the C9 protein using a PD10 column (GE Healthcare) pre-equilibrated in PBS. Fractions were collected and analyzed by UV/VIS-spectroscopy.

A total of 0.5 mg of Complement C9 (Comptech) at 1.05 mg/mL were mixed with 10 μL of 10 mM TCEP. Then, 100 ug of terbium maleimide (Life Technologies) were dissolved in 20 μL of PBS to 5 mg/mL. Then, 250 μL of the C9 solution were reacted with 2.2 μL of the terbium solution, thus at a molar dye/protein ratio of 3:1, for 2 h at room temperature. Free fluorophore was separated from the C9 protein using a PD10 column (GE Healthcare) pre-equilibrated in PBS. Fractions were collected and analyzed by UV/VIS-spectroscopy. The dye/protein ratios were 1.0 for the FITC-C9 and 0.7 for the Tb-C9.

In the HTRF assay, 2.5 μL of antibody were incubated with 2.5 μL of either 2 or 20 nM C5b6 for 60 min. Subsequently, 5 μL of a mixture of 20 nM C7, 20 nM C8, 50 nM FITC-C9 and 25 nM Tb-C9 were added and the HTRF signal read immediately using the following filter settings: Mirror = TRF D400/D505 dual/Bias, Emission Filter (1) = Invitrogen 520/25, Emission Filter (2) = Invitrogen 495/10. In the one-shot assay, the antibodies were used at a range of concentrations, with the majority being between 100–250 ug/mL FAC. For the dose response curves, antibody was diluted to the concentrations indicated and only the higher C5b6 concentration was used. To normalize the data, a control reaction with no C5b6 was used as low control and a reaction with no antibody was used as a high control. Low and high control were used to normalize the HTRF signal to 100% inhibition and 0% inhibition, respectively.

### 2.9. Liposome Leakage Assay

A liposome leakage assay was performed using a protocol adopted from [29,30]. A lipid mixture of 60/30/10% DOPC, DOPE and cholesterol (all Sigma Aldrich) was dissolved in methanol, dried under nitrogen and then resuspended in PBS supplemented with 50 mM sulforhodamine B (Sigma Aldrich) at a total concentration of 20 mg/mL. The suspension was then freeze–thawed three times and liposome size was reduced by 11 passages through an 800 nm pore size polycarbonate membrane (Avestin). Free sulforhodamine B was removed using a PD10 desalting column followed by gel filtration using a 16/60 S200 Superdex column (both GE healthcare). Both columns were pre-equilibrated in PBS. Antibody clones were incubated with 4 nM C5b6, 12 nM C8, 140 nM C9 (all Comptech) and a 1/25 dilution of liposomes diluted in PBS for 1 h in a black 384-well plate (Greiner) with 30 μL per well. Then, 10 μL of 20 nM C7 (Comptech) were added to start MAC formation and fluorescence intensity measured at 540 nm excitation and 590 nm emission. Fluorescence intensity was measured kinetically, in well-mode, every 0.5 s for 20 s and a linear slope fitted to the curve between seconds 8–13. An anti-MAC antibody was used as a positive control for liposome leakage inhibition. For the assay development, the anti-MAC (anti-C7) antibody was at 250 nM and the isotype control was at 500 nM final assay concentration, or as indicated in the figure. For the screening of the affinity matured antibodies, all antibodies were at ~1 μM.

## 3. Results

### 3.1. Dose Prediction Modelling

We aimed to generate a therapeutic antibody that would bind to a neoepitope on C5b6 and prevent C7 from binding and thus prevent MAC formation. Reported on-rates of C7 for C5b6 vary between 4 × 10^5^ and 2 × 10^6^ Ms^−1^ with no measurable off-rate and a serum concentration of approximately 500 nM [23,24]. Assuming a monthly dosing regimen of 10 mg/kg, with an overall antibody affinity of 1–2 pM and an on-rate of 1 × 10^7^ Ms^−1^ (close to the diffusion limit of IgGs), minimum target engagement was calculated to be between 81% and 93%, dependent on C7 on-rate (Figure 1C). Furthermore, the therapeutic antibody would need to bind the C5b6 neoepitope specifically and not bind to the highly abundant complement C5 and C6 proteins to avoid target-mediated drug disposition that has been reported for Eculizumab [12].

### 3.2. Naïve Antibody Library Selections

Rodent immunization was considered, since such an approach has been successful in generating monoclonal antibodies specific to human complement proteins [9]. However, due to the requirement to identify antibodies specific for C7-competitive epitopes and the fact that human C5b6 is readily neutralized by murine C7, we believed that an in vivo immunization approach would not deliver the desired antibody specificity [31]. Therefore, a yeast based in vitro selection approach was chosen in this case and antibody clones were selected from Adimab LLC platform libraries. Selections consisted of two selection cycles of multiple rounds of MACS and FACS selections. Initially, selections were performed from naïve libraries using MACS and FACS, selecting for binding to C5b6. The heavy chains from the selected clones were then cloned into a diversified light chain library from Adimab LLC and the resulting antibodies were subjected to the selections as described in Figure 2A. The antibodies were selected for C5b6 binding and a lack of binding to C5, C6 and a poly-specificity reagent. To increase the proportion of antibodies targeting C7 competitive epitopes, C7 competition was included in the selection strategy (Figure 2B).

Figure 2B describes the fifth round of selections where C7 competition was employed. Antibodies were pre-complexed with C5b6, followed by incubation with C7 and stained for C5b6 and C7 binding as well as IgG expression. Those antibody clones that could be labelled with both C5b6 and C7 were presumed not to target a C7-competitive epitope and were discarded. A FACS selection strategy was deployed to select antibodies that show strong C5b6 labelling but lack C7 labelling (Figure 2B). The C5b6 concentration was lowered 10-fold to 1 nM in the final round of selections to apply selection pressure and allow selection of the highest affinity binders. Following the final round of selections, 576 unique clones with different CDR regions were identified.

### 3.3. Characterisation of Naïve Antibody Library Selection Outputs

Full length human IgG1 molecules were expressed from yeast cells and the secreted antibodies purified using protein A affinity chromatography. Given the high affinity of the C5b6-C7 interaction, we considered it unlikely that antibodies selected without further affinity improvement would show measurable functional activity. Characterization of the antibodies thus focused primarily on antigen binding and specificity, with some functional screening included.

The antibodies were captured on a protein A chip and measured for binding to C5, C6 and C5b6 at a concentration of 60 nM using biolayer interferometry (BLI). Figure 3A shows the relative binding responses for each clone against the three antigens. While a large proportion of clones show binding to either C5 or C6, a subset shows strong binding to C5b6 and considerably weaker binding to either C5 or C6, suggesting that some of the selected clones are specific for C5b6. At this stage of the discovery process with more than 500 clones, only BLI was performed to obtain relative binding responses as opposed to absolute binding kinetics from a multi-cycle SPR experiment. As such, no conclusions could be drawn about the absolute binding kinetics or fold specificity of each clone for C5b6 over C5 or C6. A total of 31 clones with the best selectivity profiles were then tested for their ability to inhibit TCC formation in a solution phase assay. The anti-C5 control antibody significantly (*p* < 0.05) reduced TCC formation. However, whilst a reduction in TCC activity could be seen for all clones tested with some antibodies inhibiting TCC formation more strongly than the isotype controls, none of the clones from this campaign achieved statistical significance (*p* < 0.05) in this assay (Figure 3B, raw values and relevant mAb concentrations are given in the Appendix A).

Eight clones with the best binding profiles were chosen to be improved using affinity maturation via CDRH1 and CDRH2 diversification using the same yeast-based platform used for naïve selections. These eight clones show strong binding to C5b6, considerably weaker binding to either C5 or C6 (Figure 3C) and have diverse CDRH3 sequences (Figure 3D). None of the selected clones contained sequence liabilities such as unpaired cysteines or N-linked glycosylation sites.

### 3.4. Affinity Maturation of Antibody Hit Panel

As most antigen binding occurs via the heavy chain, our standard process to diversify CDR1 and CDR2 of the heavy chain was employed, with a further error prone diversification on all CDRs planned for the next round of affinity maturation if needed. The eight antibodies chosen for affinity maturation were diversified in CDR1 and CDR2 of the heavy chain using the Adimab LLC platform. Library sizes of approximately 2 × 10^8^ clones were generated and subjected to C5b6 selection as per the naïve selections with the addition of competition from the parental IgGs in the final round to sort for clones with higher affinities than parental clone. Of the eight libraries, six contained binders that had increased binding affinities as observed in FACS plots (Appendix A). Approximately 0.1% of the population that had higher affinity than parental clones were selected in the final round of FACS. A total of 760 clones were sequenced from the eight libraries, yielding 349 unique clones.

### 3.5. Development of Antigen Binding and Complement Function Assays to Screen Affinity Matured Antibodies

#### 3.5.1. Development of a Surface Plasmon Resonance Assay Format Suitable for C5b6

A surface plasmon resonance (SPR) assay was developed to determine the binding kinetics of the anti-C5b6 antibodies. While measurement of the binding response by biolayer interferometry was sufficient to rank clones from the first antibody selection rounds, accurate binding kinetics were required during the later stages of characterization. In particular, as the dose–response modelling highlighted the requirement for a fast on-rate. In addition, a reliable SPR assay enables comparison of the antibody affinities before and after affinity maturation and thus the success of the maturation process can be evaluated.

A standard SPR assay using captured antibodies as ligands and antigen as analyte was not suitable as C5b6 bound non-specifically to SPR chip surfaces. To overcome this challenge, C5b6 was captured on an immobilized anti-C5 antibody and the antibodies to be characterized were used as analytes (Figure 4A). This assay format removed C5b6 from the solution phase during analysis cycles to minimize the effect of non-specific C5b6 interactions and enabled multi-cycle kinetic experiments to be performed via regeneration of the anti-C5 antibody at low pH.

To confirm the captured C5b6 was still in its active form, the entire TCC was assembled on the SPR chip. Either C5b6 or C5 were captured on the anti-C5 antibody and C7, C8 and C9 were injected in succession. As expected, the three analytes showed no binding to C5 (Figure 4B inset), but strong, virtually irreversible binding to C5b6 that required the successive addition of C7, C8 and C9 to the captured C5b6 (Figure 4B main). Exclusion of single components precluded binding of the subsequent components. The on-rate of C7 binding to C5b6 was measured using this assay format (Figure 4C). The observed on-rate of 2.3 × 10^6^ Ms^−1^ agrees closely with the 2.0 × 10^6^ Ms^−1^ reported previously in a similar assay format using an anti-C6 antibody [24]. These data suggest that trough target engagement would likely be at the lower end of our simulation at approximately 80%. The TCC assembly on the chip suggested that C5b6 captured by the anti-C5 antibody does not occlude any C7 binding sites, i.e., the neoepitopes of interest. Mapping the epitope of the anti-C5 mAb onto the structure of full membrane attack complex also suggests antibody binding should not interfere with TCC assembly (Appendix A). For clones whose epitope overlaps with the anti-C5 capture antibody, this assay set up may mask binding to C5b6 and particularly C5, but we deemed this risk acceptable as such clones would likely not be C7-competitive, judged by the SPR experiment in Figure 4B and the structural modelling in Appendix A, and would be removed in the screening cascade elsewhere.

Subsequently, we assessed the binding affinities of the eight clones chosen for affinity maturation as analytes and obtained sensorgrams that fit to a 1:1 interaction model with some deviations. The fitted dissociation constants were spread over a wide range from 26 to 1500 nM (Figure 4D). As the measured off-rates of some interactions did not agree well with the fitted curves, most likely due to the avid nature of the assay using bivalent IgGs as analytes, the overall calculated affinities may not be accurate due to the avidity effect arising from cross-linking of the two chip surface antigens by the two mAb binding sites from each Fab arm [32].

To determine if this assay format could be used to determine absolute binding kinetics with good accuracy, Fab fragments of two of the clones were tested in the assay (Figure 4D right). The interactions of the monovalent Fab fragments show good agreement with a 1:1 binding model for both on-rate and off-rate. For affinity matured clones with high affinities, where Fabs do not require injection at higher concentrations, the large bulk effect observed here would be expected to be smaller thus improving assay data quality further. Overall, these data suggest that an assay format using an anti-C5 capture method can be used to determine the absolute binding kinetics for C5b6-antibody interactions, provided that Fab fragments are used.

#### 3.5.2. Development of Functional Complement Assays

In addition to an SPR-based antigen binding assay, two assays were developed to measure terminal complement activity in vitro. One assay measured the formation of soluble MAC and a complementary assay measured the formation of lytic MAC on phospholipid membranes. The assays were designed to be more sensitive to MAC inhibition than a hemolysis assay in order to identify C7-competitive antibodies with low affinity. Both assays used purified complement proteins and involved a pre-incubation step of C5b6 with the antibodies to allow C5b6-antibody complex formation, thus increasing the probability of identifying clones with slow on-rates that could be improved by affinity maturation.

The relative protein concentrations in the assays were optimized to make the MAC assembly limiting for C5b6 and therefore further increase assay sensitivity. The two assay formats described below were chosen to increase the chances of finding functionally relevant clones and to account for any differences between soluble MAC and MAC bound to a membrane.

In addition to the soluble MAC assay, a fluorescent assay for the lytic activity of MAC was developed based on the work of Faudry et al. [29]. Liposomes encapsulating sulforhodamine B fluorophore, at a self-quenching concentration of 50 mM, were pre-incubated with C5b6, C8, C9 and antibodies. Following addition of C7, MAC is formed, the liposomes burst and dilution of the fluorophore into the surrounding buffer leads to an increase in fluorescence intensity. (Figure 5A). The data show that upon addition of C7, MAC is formed, and liposomes burst in a process which takes approximately 60 s to complete. An anti-MAC control antibody was shown to provide protection from lysis in a specific (Figure 5B) and dose-dependent (Figure 5C) manner, enabling determination of IC50 values.

Soluble MAC formation was measured via a Homogeneous Time Resolved Fluorescence (HTRF) assay where the antibodies were pre-incubated with C5b6 before addition of C7, C8, and C9 labelled with a FRET donor and C9 labelled with a FRET acceptor. Upon MAC formation, the previously monomeric C9 oligomerizes (Figure 1A), leading to an increase in HTRF signal.

The two functional assays and the antigen binding assay described here enabled sensitive screening of the affinity matured antibodies for binding affinity, kinetics and potency.

### 3.6. Characterisation of the Affinity Matured Antibody Output and Comparison with Parental Antibodies

#### 3.6.1. Antigen Binding Properties of Affinity Matured Antibodies

The affinity matured antibodies were expressed, purified and analysed in order to determine which of the diversified clones showed an improvement in affinity, selectivity and potency. As standard, prior to expression and purification the sequences of the antibodies were analysed for predicted stability liabilities. Once purified, the biophysical characteristics of the antibodies were assessed to ensure there were no stability issues that were not picked up in the initial sequence liability assessment. Techniques used included an in silico sequence liability check, analytical size exclusion chromatography and liquid chromatography mass spectrometry.

As before, the clones were triaged initially by their binding to C5, C6 and C5b6. For C5 and C5b6 binding, the SPR anti-C5 capture assay described in Figure 4 was used (Figure 6A). Binding of the antibodies to C6 was tested using the same BLI assay as described in Figure 3A, (Figure 6A). At this stage of the discovery process, more stringent selection conditions were desired and the concentrations of the C5 and C6 analytes in the assays were increased to a more physiologically relevant concentration of 500 nM, approximating their serum concentrations. Under these conditions, a larger proportion of antibodies were identified with residual binding to C5 or C6. In the triaging process, only binding responses of full-length IgG1 molecules were measured, with the intent to generate Fab fragments from a subset of the most improved binders to determine accurate binding kinetics.

The binding data from either set of experiments showed that while some clones are neoepitope specific, most of the affinity matured antibodies also bind either C5 or C6 (Figure 6A). More precisely, it was shown that the residual binding to C5 or C6 is lineage specific. Lineages A–D, derived from clones A–D in the panel of initial hits, showed residual binding to C5, while lineages E–H showed residual binding to C6. Even when selective pressure against C5 and C6 binding was applied, parental and daughter clones showed similar patterns of residual non neo-epitope binding. As could be expected, the introduction of point mutations in the CDRH1 and CDRH2 regions did not alter the overall specificities of the antibodies.

In the case of two lineages however, the binding data were more encouraging and showed that the process had selected a small number of daughter molecules with superior affinity and/or selectivity for C5b6. Several clones in lineage A showed only low binding to C5, while simultaneously displaying improved binding to C5b6 compared to the parental (first clone, marked with an arrow). Similarly, lineage E contained several antibodies that have improved binding to C5b6, while the binding to C6 was not increased. A small number of clones were identified with improved binding to C5b6 and minimal binding to either C5 or C6. The affinity maturation output was further characterized using functional complement assays before selecting clones for Fab generation and accurate determination of binding kinetics.

#### 3.6.2. Functional Characterisation of Affinity Matured Antibodies

Initially, the selection output of optimized clones was screened using the more physiologically relevant liposome leakage assay. None of the antibodies showed any functional activity in this assay.

The antibodies were then tested in the HTRF assay at two different C5b6 concentrations (Figure 6B). There is good correlation between the two assays using either 0.5 or 5 nM C5b6 and several affinity-matured clones (small circles) inhibit C9-oligomerisation more effectively than their parental clones (large circles). Where there is inhibition below 0%, we believe this to be due to noise in the assay rather than stimulation of MAC formation by the antibodies. To determine whether this apparent improvement in the single dose assay was reproducible, a subset of promising clones (Figure 6B, right) was tested in full dose response experiments (Figure 6C). Upon inspection of the dose response curves, separated by lineage with each parental clone highlighted in red, no improvement in potency is seen for the affinity matured clones compared to their respective parental clone. Although there is an increase in affinity observed in Figure 6C in lineage C, this did not confer enough of an increase in potency to be observed in the hemolysis assay. Likely due to the de novo formation of C5b6 and presence of physiological ratios of C5b6 and C7.

While the liposome leakage assay is likely more physiologically relevant, the C9 oligomerization appears to be more sensitive, as both some parental and affinity matured clones showed some effect in this assay. Alternatively, the eight antibodies may be specific for soluble C5b6 only.

In summary, the functional analyses show that, although the optimized clones showed were selected for increased binding affinity using FACS, the optimization process did not increase the potency of the antibodies and that none of the antibodies from this campaign could neutralize membrane-bound MAC assembly and stop membrane pore formation. This highlights the technical challenges to select functional MAC inhibitors via targeting of neo-epitopes. Given the clear lack of improvement in terms of affinity or potency, antibody discovery efforts using C5b6 were therefore halted.

## 4. Discussion

The data presented here illustrate the challenges inherent in antibody discovery targeting neoepitopes and possible strategies to overcome these. Although we were able to select specific neo-epitope binders, we were ultimately unsuccessful in isolating an antibody with the required potency, and several lessons have been learned which may guide future antibody discovery campaigns targeting neoepitopes on challenging antigens. Previous examples of antibody discovery approaches for complement neoepitopes include phage display (S77) or mouse immunization (H17) [16,33]. More recently, a study by Zelek and Morgan (2020), provided validation of the lower dosing implications of targeting an epitope of an active intermediate terminal pathway complex (C5b7) with identification of an antibody using an in vivo approach [34].

The in vitro selections approach we used enabled us to tailor the selection strategy towards desired epitopes and specificities. By including negative rounds of selection on the FACS using both C5 and C6, we successfully identified antibodies selective for C5b6 neoepitopes with only low level C5 or C6 binding. The C7 competition in another selection round enabled us to select antibodies binding to C7-competitive epitopes. This was confirmed by functional data from the HTRF assay which shows that a subset of clones was C7-competitive. In addition, the SPR data of the affinity matured antibodies shows increased, specific binding to C5b6 over the parental clones, suggesting that the selection method deployed selects for neoepitopes on C5b6.

One of our criteria was that the selected antibodies needed to be able to outcompete the high affinity physiological binding partner C7, which is found at high concentrations in blood. Initially, a kinetic haemolysis assay using carefully titrated purified terminal pathway components (C5b6, C7, C8, C9) was employed as a primary functional screen. However, as this did not provide robust functional data on the clones, we considered the possibility that inhibition of antibodies from the naïve selection rounds may not be measurable in a haemolysis assay due to their low affinity, despite skewing of the assay condition in favour of C5b6 inhibition. Indeed, no functional inhibitors were identified in haemolysis assays using whole serum as a source of complement. Possible reasons for this are that (1) clones from naïve selection may be too low affinity to outcompete binding of C7 to C5b6 which binds with an on rate of 2.3 × 10^6^ Ms^−1^, and (2) the sensitivity of this assay type is insufficient. Therefore, we designed two additional functional assays to measure the potency of our antibodies.

A solution-based assay and a membrane-based assay were developed to measure the engagement of our antibodies to C5b6 in solution and to the slightly lipophilic C5b6 when bound to a phospholipid membrane, respectively. In the haemolysis assay using serum, the C5b6 is generated in situ, requiring the antibodies to compete with C7 in real time. The assays described here used purified terminal complement proteins, allowing for pre-incubation of C5b6 with the antibodies to increase the sensitivity of detection of antibodies that outcompete C7. We showed that both solution and membrane-based assays were able to measure the in vitro potency of antibodies targeting MAC, but the antibodies generated here only showed functional activity in the solution-based assay. This may be due to different assay sensitivities or the fact that membrane-bound C5b6 presents slightly different epitopes compared to C5b6 in solution, but it confirms that the epitopes targeted were C7 competitive.

Another challenge was the propensity of C5b6 to aggregate and bind surfaces non-specifically. Identifying and selecting the highest affinity clones by FACS was made highly challenging by this behaviour. Consequently, stored and potentially non-monomeric C5b6 may display different epitopes compared to C5b6 generated de novo in serum-based assays.

We generated a bespoke SPR assay to determine the binding kinetics of our selected clones. C5b6 was captured using an anti-C5 tool antibody and candidate antibodies were used as analytes. Fab fragments were used instead of full length IgGs and the sensorgrams produced agreed with a standard 1:1 binding model and could be used to determine absolute binding kinetics. Using a tool antibody to capture a challenging antigen on an SPR surface is a promising method when working with antigens that are aggregation prone and cannot easily be regenerated, as long as the tool antibody does not occlude the desired epitopes. The measured dissociation constants ranged between 26–1500 nM, several orders of magnitude below the desired target affinity of 1–2 pM. Similarly, the measured on-rates were lower than the desired on-rate of 1 × 10^7^ Ms^−1^ and did not show the desired potency. Some affinity matured clones showed stronger binding by BLI, but this did not translate to improved activity in functional assays, most likely due to the higher sensitivity and precisions of the BLI assay.

Overall, the data demonstrate the utility of a yeast-based antibody platform in tailoring selection strategies to required specificities and selectivity. We also highlight the value of bespoke assay formats and reagents to characterize challenging drug targets in light of the inherent difficulties in an antibody discovery campaign aiming to deliver high-affinity neoepitope-specific molecules. An antibody was sought that was neoepitope-specific, binds a C7 competitive epitope and shows high-affinity for its target. This work highlights the challenges of discovering an antibody that possesses all these characteristics and provides valuable lessons for antibody discovery against targets in the complement system.

## Figures and Tables

**Figure 1 antibodies-10-00039-f001:**
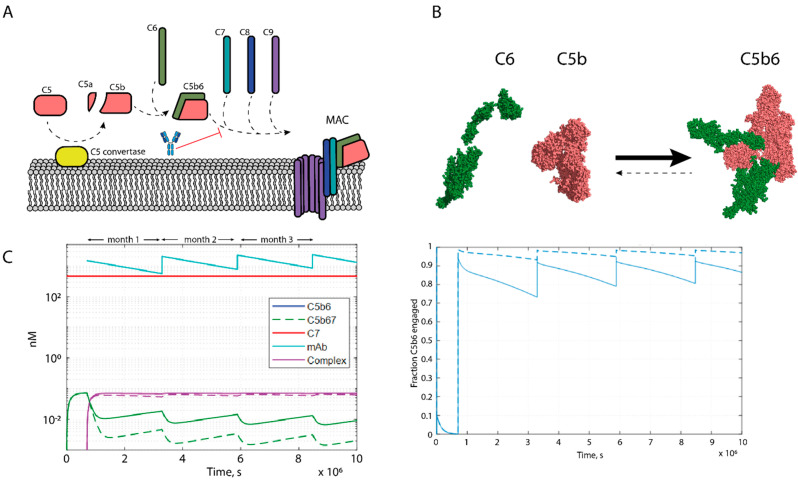
Schematics of biology, desired molecule and antibody selection strategy. (**A**) Schematic representation of membrane attack complex formation, indicating the interaction targeted by the proposed therapeutic. (**B**) Space filling representations of the atomic models of C6 (PDB: 3T5O), C5 (PDB: 3CU7) and C5b6 (PDB: 4A5W), illustrating the formation of neoepitopes. (**C**) Dose prediction modelling of an anti-C5b6 antibody with 1 pM affinity and an on-rate of 1 × 10^7^ Ms^−1^ and onthly dosing of 10 mg/kg. Simulation carried out for both reported C7−C5b6 on-rates; 4 × 10^5^ Ms^−1^ (dashed line) and 2 × 10^6^ Ms^−1^ (solid line). Fraction target engagement plotted on the right.

**Figure 2 antibodies-10-00039-f002:**
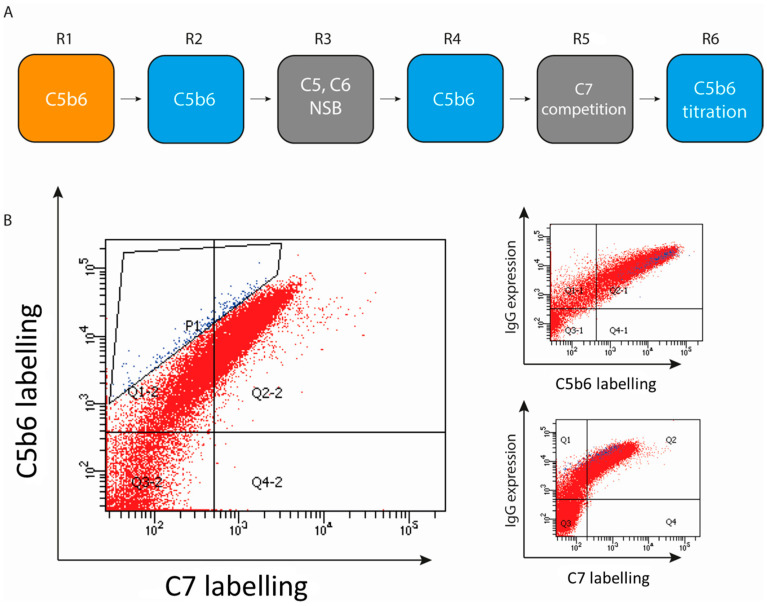
Naïve selections strategy and sorting. (**A**) Flow diagram of the 6 rounds of selection to isolate anti-C5b6 antibodies from a yeast library. Selection rounds are colour coded as MACS (orange), FACS (blue) and negative selection FACS (grey). NSB refers to a proprietary reagent to select against antibodies displaying nonspecific binding. (**B**) FACS plots of round 5, where clones were selected that show strong binding to C5b6, but no binding to C7 once bound to C5b6 already. These are coloured in blue. All three plots show the same population, with different parameters plotted on each axis.

**Figure 3 antibodies-10-00039-f003:**
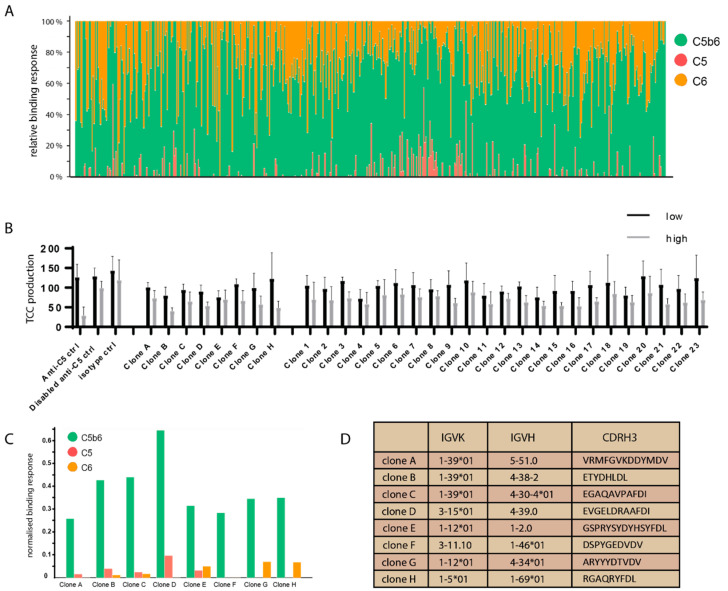
Characterisation of naïve selections output (**A**) Relative binding responses measured by BLI of the antibodies from light chain batch shuffle against three antigens, C5 (red), C6 (amber), C5b6 (green) at 50 nM. Stacked bars with height indicate relative binding responses for each of the 3 analytes. Each bar represents one clone. (**B**) Level of terminal complement complex production in the presence of low (black) and high (grey) antibody concentrations, visualised in a bar chart. The output was tested in 1% NHS at ~1–2 μM (high), 0.02–0.06 μM, the controls were added as follows: anti-C5 and disabled anti-C5 (0.02/0.01 μM), isotype control at 14/0.44 μM. (**C**) BLI binding responses from (**A**) filtered to the 8 clones that were chosen as a hit panel. Side-by-side bars indicating the respective, absolute binding responses. Data in A and C has been normalised for the differences in molecular weight of the analytes. (**D**) Antibody heavy chain and kappa light chain genes and CDRH3 sequences.

**Figure 4 antibodies-10-00039-f004:**
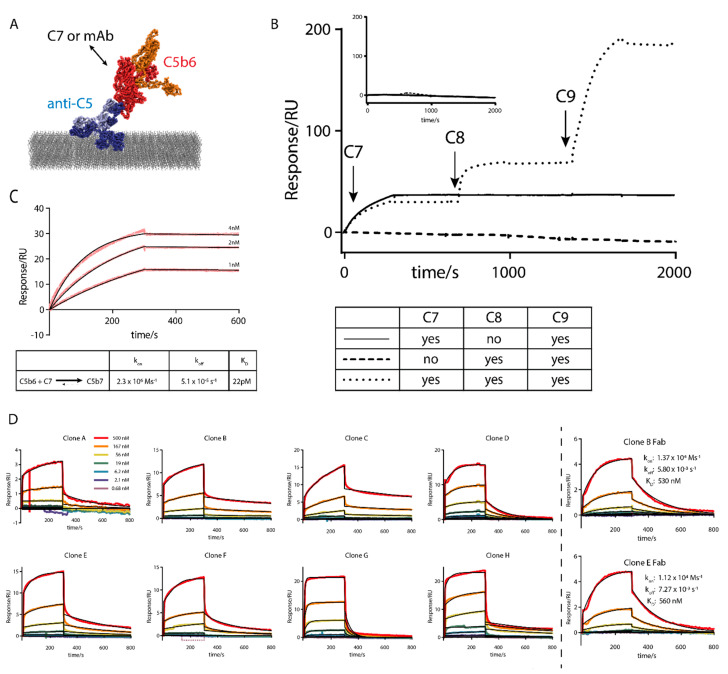
MAC on a chip and a novel SPR assay format. (**A**) Schematic representation of the SPR assay format. Composite model generated using PDB depositions 5I5K and 4A5W. An anti-C5 tool antibody (blue) is immobilised on a CM5 chip (grey) and C5b6 (red/orange) is captured by the antibody. Analytes can then be tested for binding against C5b6. (**B**) Reconstitution of the membrane attack complex on an SPR chip surface. Analytes injected as indicated by the legend. Experiment was performed with either captured C5 (inset) or C5b6 (main). (**C**) Measuring the on−rate of C7 on C5b6. Experimental data in red and fit to a 1:1 binding model in black. Table of kinetic parameters below. (**D**) Binding of the 8 hit panel clones to anti-C5 captured C5b6. Experimental data coloured as indicated in the legend. Fits to a 1:1 binding model in black. Excluded data shown as dotted lines. Binding kinetics only reported for the true 1:1 Fab-binding.

**Figure 5 antibodies-10-00039-f005:**
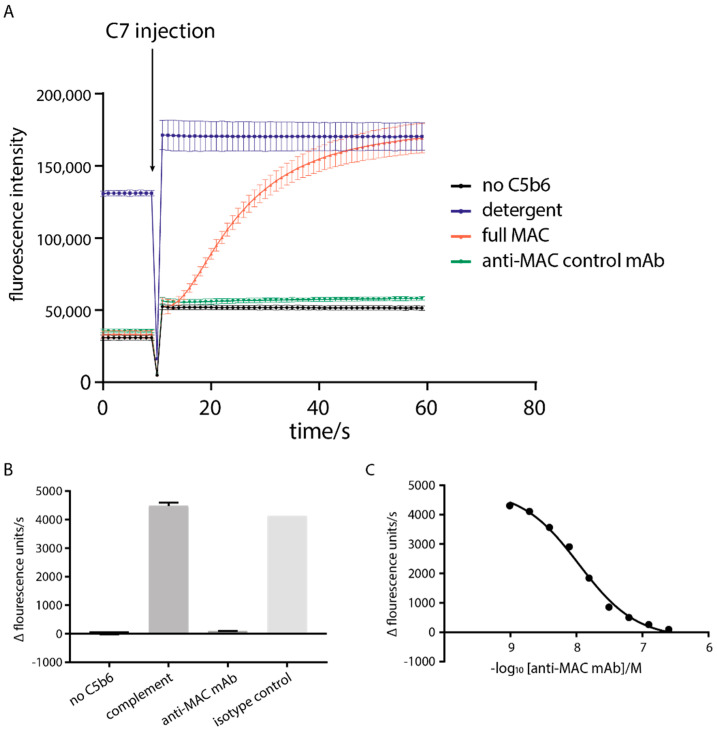
Development of functional complement assays. (**A**) Liposome leakage assay development. Fluorescence intensity measured kinetically and C7 injection time point indicated. The assay was run using full MAC (red), a no-C5b6 low control (black), full MAC plus detergent high control (blue) and full MAC plus an anti-MAC tool antibody (green). (**B**) Liposome assay development using an anti-C7 mAb was used anti-MAC positive control antibody and an isotype control antibody. (**C**) Dose−dependent effect of the anti−MAC antibody on the kinetics of liposome formation.

**Figure 6 antibodies-10-00039-f006:**
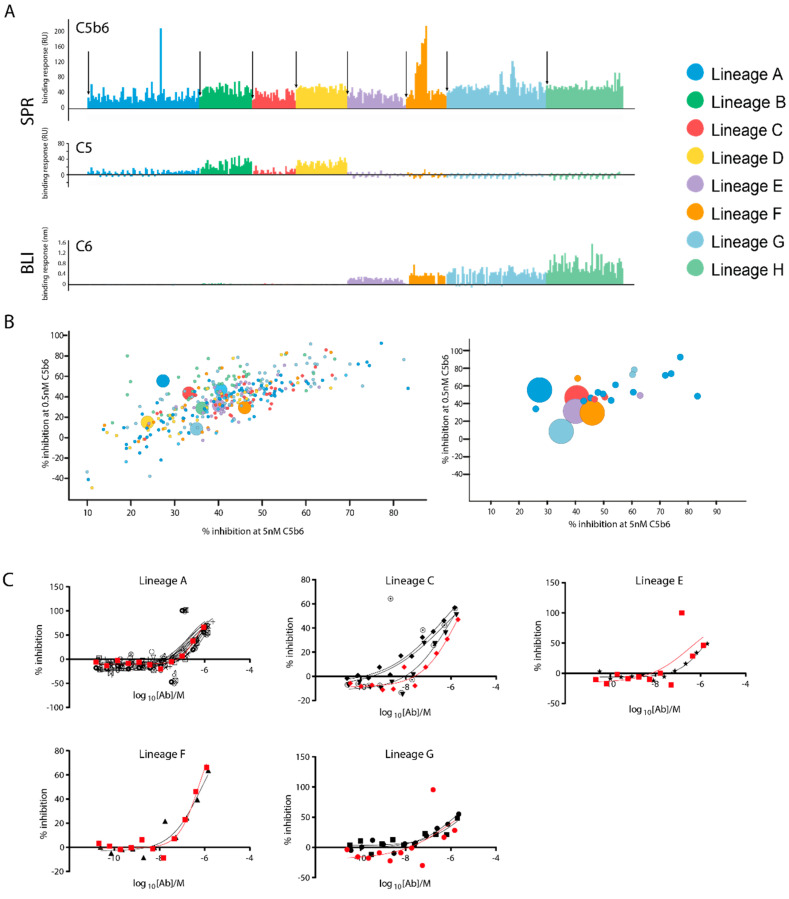
Characterisation of the affinity matured output. (**A**) Relative binding responses of affinity matured clones for the 3 antigens. C5 and C5b6 binding is measured by SPR using the capture method described in Figure 4, binding to C6 is measured by BLI. Clones are coloured by lineage and the parental clone, the first bar for each lineage, is highlighted with an arrow. The C5 and C5b6 binding responses are plotted on the same axis and not normalised for the relative molecular weights of C5 and C5b6. (**B**) %−inhibition in the C9-oligomerisation assay with 0.5 nM C5b6 and 2.5 nM C7 (*y*−axis) and 5 nM C5b6 and 10 nM C7 (*x*−axis). Each circle represents one clone, coloured by lineage. Parental clones are shown as larger circles. Right: A selection of 24 promising clones, from 5 lineages, to be further analysed in a dose−response HTRF assay. (**C**) Dose response curves of the 24 clones tested in the HTRF assay using 5 nM C5b6 and 10 nM C7, grouped by lineage, with parental clones highlighted in red.

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
