# Peer review of "Novel Selection Approaches to Identify Antibodies Targeting Neoepitopes on the C5b6 Intermediate Complex to Inhibit Membrane Attack Complex Formation"

_2073-4468, 2021, doi:10.3390/antib10040039_

Round 1
Reviewer 1 Report
This is a very interesting study. My specific comments are:
- Line 137 the sentence “In round 4, C7 competition using unlabeled C5b6 was performed.” seems redundant. The sentence(s) right before carries the information already.
- Line 287 “a subset shows strong binding to C5b6 and considerably weaker binding to either C5 or C6”: As C5 and C6 are much more abundant compared to the intermediate C5b6 in physiological environment, I wonder how weak the binding to C5/C6 should be desired? Perhaps, simulation can provide some additional insights. Also, for these isolated clones, their affinity measurements on C5/C6 were missing.
- Date of Fig 3B, does it suggest that anti-C5 control was simply better than all characterized clones in term of TCC inhibition? If so, will the central hypothesis on producing C5b6 specific mAbs become questionable? Or just because the potency was not good enough for the clones of this study? By the way, more spec on this anti-C5 control (e.g. its origin and affinity to C5) could be introduced to help readers better understand.
- To claim “specific for neoepitopes” (line 289), I feel at minimal some epitope mapping, competitive ELISA / sequential SPR should be performed as direct evidence to back up the claim.
- Date of Fig 3C, is that one point ELISA? It was not clear, and conditions were missing as well.
- Line 307 “Of the eight libraries, six contained binders that had increased binding affinities as observed in FACS plots” important FACS data not shown unfortunately. Or SPR/Octet data of affinity maturated clones will be valuable as well.
- What was the anti-MAC control antibody used in Fig 5? Description with citation (if appliable) will be useful.
- Line 395: “The affinity matured antibodies were expressed, purified and analyzed to determine which of the diversified clones showed an improvement in affinity, selectivity and potency.” No date shown to support the statement.
- Considering C7 presents at high concentration of 460 nM and has a very high affinity of 2pM to C5b6, anti-C5b6 mAbs of 1-10 pM Kd will be needed to compete C7 (as simulation suggests). However, the clones A-H only showed nanomolar affinities. Therefore, not a surprise that these mAbs failed in membrane-based assay of MAC inhibition. What were the exact potencies of affinity matured clones?
Reviewer 2 Report
The authors present the results of an antibody campaign designed to generate binders to a neoepitope formed upon complexation of complement components C5b and C6. The yeast display strategy incorporated multiple rounds of positive selection for C5b6 binding and negative selection for binding of C5, C6, a non-specificity reagent, and C7 binding following C5b6 complexation. While neoepitope-specific antibodies were generated, they generally had weak affinity for C5b6 (all >20 nM KD) and residual binding to C5 or C6. Subsequent affinity maturation led to some increases in C5b6 binding and specificity, as well as activity in a solution-based MAC inhibition assay. However, lack of functionality in a more relevant liposome leakage assay suggested there would be no biological function of preventing pathogenic MAC formation on cells.
Although the presented antibody campaign was not successful in generating clinical candidates, the authors take the opportunity to highlight lessons learned through the process. Thus, the manuscript is potentially of interest and utility for other scientists attempting to generate antibodies against neoepitopes created upon association of protein binding partners.
Major points:
- The abstract states that the antibodies “demonstrated competitive binding with C7.” While it is true that antibodies were selected in part based on low binding to C7 after complexation with C5b6, purified antibodies were never used in competitive binding assay, for example SPR or BLI. Thus, I think this language in the abstract should be reworded, unless the authors have more data explicitly showing competitive binding.
- It should be emphasized that the SPR assay using anti-C5 to capture C5b6 measures avidity, and is therefore likely to overestimate the strength of the measured interaction. It should also be noted that, consistent with this notion, the measured affinity for the two Fabs was ~10-fold weaker than the measurement for the corresponding mAbs.
- In BLI/SPR figures (especially figures 3A, 3C but also 6A) showing binding response of C5, C6, and C5b6, the responses should be normalized by molecular weight in order to fairly compare the binding to each protein. I appreciate the note to clarify that responses were not normalized, but I think normalization is important since currently the experimental reporting is biased toward “desired” binding to the complex, simply because it is larger and will cause a larger binding response. Alternatively, more discussion could be added in the text to address this inherent bias.
Minor points:
- Could the authors briefly explain the rationale for using a wild-type IgG1 backbone containing active effector functions? The approved C5 antibody is IgG2/4, which helps to decrease ADCC, ADCP, and CDC that are not required for a mechanism of strict inhibition. On the other hand, use of IgG1 subtype allows binding to Fc gamma receptors, which could be detrimental for safety and/or PK.
- Antibodies were purified by protein A and subsequently used for binding and functional assays. Without selection for stability, there is a chance that some antibodies could have significant aggregation. Was protein quality ever assessed, and is there any reason to think that aggregation could impact downstream assays?
- Has the epitope of the C5 antibody been identified? The authors point out that successful assembly of MAC on the SPR chip suggests that the sites on C5b6 for anti-C5 binding and C7 binding do not entirely overlap, but structural data could support this claim.
- For BLI experiments, due to fast dissociation rate proA sensors are most useful for titer quantitation rather than binding evaluation. Just a friendly suggestion to try AHC sensors for antibody kinetics in the future.
- Could the authors briefly explain why affinity maturation was performed by altering CDRs H1 and H2 specifically?
- Do the authors have information on the anti-MAC mAb and its mechanism of inhibiting MAC formation?
- It should be noted that use of the anti-C5 capture method for the SPR assay with affinity-matured antibodies could result in lower C5 binding as an artifact of the assay setup. This could reduce apparent “non-specific” C5 binding response.
- Although the authors state that affinity maturation did not appear to increase potency, some of the affinity-matured clones in lineage C appear to be more potent than the parental clone (figure 6C). Also, can the authors speculate why increased C5b6 affinity and specificity did not translate to increased potency, for example in the successful lineages A and E?
- I saw the note in the text, but it would be helpful to point out within the figure legend that figure 4B (main) shows C5b6 while 4B (inset) shows C5.
- How do the authors interpret the increase in MAC formation caused by some antibodies in figure 6B? Could binding to some C5b6 epitopes actually increase MAC formation, or does this represent noise in the assay?
Round 2
Reviewer 1 Report
All my review comments are addressed.